

# A new borehole electromagnetic receiver developed for CSEM methods

Sixuan Song[1], Ming Deng[1], Kai Chen[1], Muer A[2], Sheng Jin[1]

[1]School of Geophysics and Information Technology, China University of Geosciences (Beijing), Beijing, China

5    [2]Hangzhou Hikvision Digital Technology Co.,Ltd., Hangzhou, China

*Correspondence to*: Ming Deng (dengming@cugb.edu.cn)

**Abstract.** Conventional surface electromagnetic methods have limitations of shallow detection depth and low resolution. In an attempt to increase the detection depth and resolution, borehole-surface electromagnetic methods for electromagnetic three-dimensional observations of ground, tunnels, and boreholes has been developed. Current borehole receivers only measure a single parameter of the magnetic field component, which does not meet the special requirements of controlled source electromagnetic (CSEM) methods. This study proposes a borehole electromagnetic receiver which realizes synchronous acquisition of the vertical electric field component in the borehole and the three-axis orthogonal magnetic field components. This receiver uses Ti electrodes and fluxgates as sensors to acquire electric and magnetic field components. Multi-component comprehensive observation methods that add the electric field component can effectively support the CSEM method, improve detection accuracy, and show broad potentials for detecting deep ore bodies. We conducted laboratory and field experiments to verify the performance of our new borehole electromagnetic receiver. The receiver achieved magnetic field noise less than 6 pT/√Hz at 1 kHz, and the electric field noise floor was approximately 10 nV/√Hz at 1 kHz. The -3 dB electric field bandwidth can reach DC~10 kHz. Results of our experiments support the claim that high-quality CSEM signals can be obtained using this new borehole electromagnetic receiver, and that the electric field component exhibits sufficient advantages for measuring the vertical component of the electric field.

Keywords:Borehole EMR, Borehole-surface electromagnetic method, CSEM

## 1 Introduction

The borehole-surface electromagnetic method is an electromagnetic survey method that supplies a high-power alternating current with a horizontal electrical dipole and receives an electromagnetic response from the ground, tunnel, or borehole being measured. In comparison with the conventional surface electromagnetic method, the borehole-surface electromagnetic method has a deeper detection depth and a higher resolution. With the expansion of research on borehole-ground electromagnetic methods, borehole electromagnetic instrumentation has developed rapidly. A variety of borehole-ground electromagnetic receivers have been developed for different methods. The DigiAtlantis Probe, produced by EMIT (Duncan et al., 1998), has a full waveform recording function based on three-channel magnetic field



measurement, and is equipped with numerical simulation software. Crone's PEM system (Crone physics, 2018) calculates the pulse step response from the dB/dt curve to extract weak anomaly information from the borehole and realizes a 157 dB large dynamic range for three-axis magnetic measurements.

Most of the receivers currently in use only measure magnetic field components. However, controlled source electromagnetic (CSEM) methods involves multi-component measurement of both electrical and magnetic signals.
Multi-component data can help researchers better interpret the relevant properties of subsurface media. Therefore, it is necessary to develop a borehole electromagnetic detection system that acquires electrical and magnetic signals simultaneously. Then the inversion results of the observed data are combined with logging data to improve the interpretation accuracy, so as to better observe geological anomalies and provide comprehensive analyses useful for minerals, oil and gas, and engineering exploration.

This paper focused on the development of a borehole electromagnetic receiver (borehole EMR) aimed at acquiring electromagnetic multi-component measurements in boreholes. It realizes high-precision acquisition of the three-axis magnetic field components and the vertical electric field component in the borehole, with broad bandwidth and large dynamic ranges, and stores and transmits status data that contains the root mean squares (RMS) of the magnetic and electric field signals, attitude, orientation, depth, and temperature. These data are sent to the wellhead unit.

## 2 Hardware development

As shown in Figure 1, the new borehole EMR includes an aluminum alloy probe tube, cable, and wellhead unit. The electronic circuit inside the probe tube includes an electrode, fluxgate, acquisition circuit, control circuit, attitude circuit, communication circuit, and a lithium battery, from bottom to top. The sensitive fluxgate and electrodes are located at the bottom of the unit, which is designed to minimize electromagnetic interference between the modules.

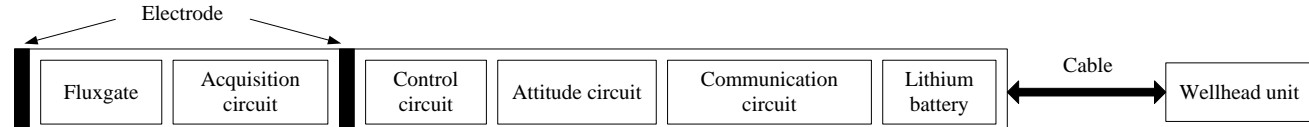

**Figure 1: Structure of the hardware in the borehole EMR.**

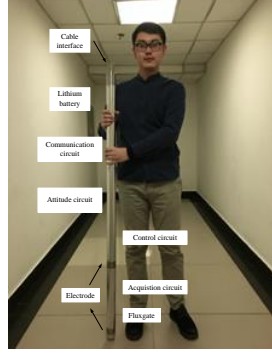



**Figure 2: Picture of the borehole EMR. Probe top and cable connections communicate with the wellhead unit.**

Figure 3 shows a block diagram of the internal hardware architecture of the borehole EMR. The fluxgate performs

high-precision conversion of the three-axis orthogonal magnetic field to the voltage signal, and the voltage is output to the subsequent acquisition circuit. The three-channel voltage signal output from the fluxgate and the voltage signal output from the electrode are then subjected to low-noise amplification, filtering, and analog-to-digital conversion through an acquisition circuit. The attitude circuit includes temperature sensors, accelerometers, and other modules to perform basic status parameter measurements. We used an acorn RISC machine (ARM) and field programmable gate

array (FPGA) together as the main control solution to complete the functions of the unit: this reads the temperature information, attitude information, and electrical and magnetic information output from the acquisition circuit, and performs pre-processing to form a data packet. The data packet is then sent to the communication circuit under the control of the wellhead unit. The communication circuit uses full-duplex communication between the probe tube and the wellhead unit. The time synchronization circuit uses a global positioning system (GPS) and oven controlled crystal

oscillator (OCXO) for time synchronization between the borehole EMRs and the transmitter. The lithium battery supplies power to the borehole EMR. It should be noted that in order to reduce power supply interference and the weight of the armored cable, the cable is only capable of simple communication and has no power supply function. The capacity of a single lithium battery in the unit is 2.5 Ah. The lithium battery pack is made up of three sections in series that can provide continuous power for nearly 24 hours.

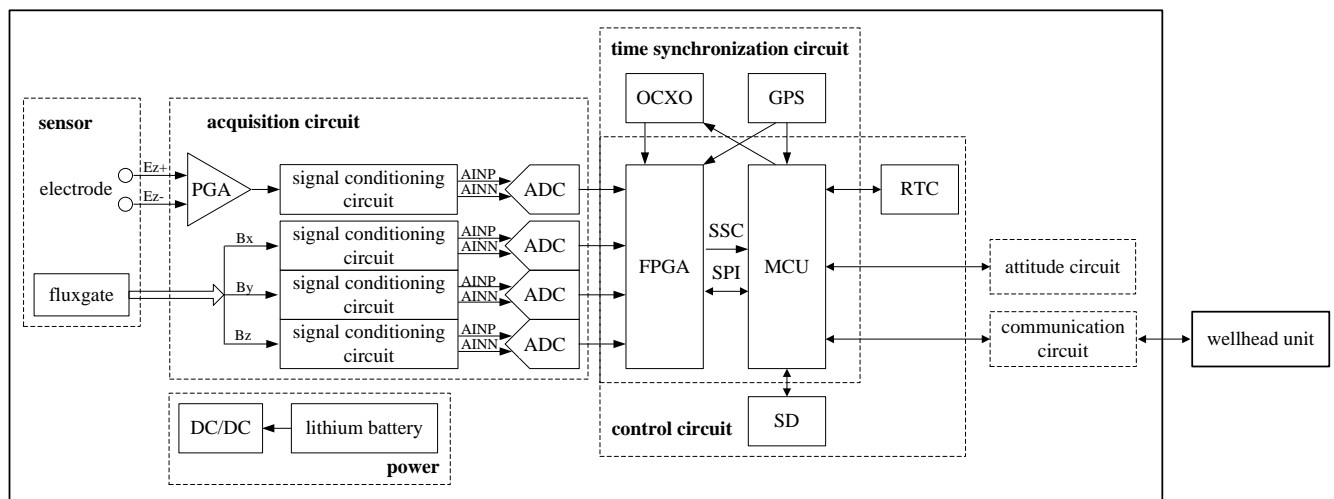

**Figure 3: Block diagram of the internal hardware architecture of the borehole EMR. The MCU here refers to AT91SAM9G45.**

## 2.1 Sensor

### 2.1.1 Electrode

Studies have shown that Ti electrodes and metal salt electrodes, such as $Pb/PbCl_2$ and $CuSO_4$ electrodes, are often

used for terrestrial observations of electric fields. The $Pb/PbCl_2$ and $CuSO_4$ electrodes have stable polarization potential differences and good consistency (Petiau, 2000). However, a borehole environment has high temperatures, high

pressures, and strong corrosion. In order for the signal-to-noise ratio of the signal observation to be within the measurement requirements, we chose a Ti electrode with a small potential difference and stability as the electric field sensor. In addition, the electrode has high strength, high hardness, high temperature resistance, and strong corrosion resistance, which are important for adapting to the borehole environment. The electrode is integrated into the borehole EMR and is mounted at the bottom of the borehole EMR unit. Its size is small, with an outer diameter of 50 mm and a height of 30 mm. The dipole length between the two electrodes is 50 cm.

### 2.1.2 Magnetometer

For measuring the magnetic field, fluxgates, induction coils, Optical Pump Magnetometers, Proton Magnetometers, superconducting quantum interference devices (SQUID), Giant Magneto-resistance (GMR), Giant Magneto-impedance (GMI), etc. are often used. We used a fluxgate sensor that is currently one of the best choices for field applications. The fluxgate sensor is more robust than an induction coil sensor for borehole observations made while in motion. The measuring object of the Optical Pump Magnetometer and Proton Magnetometer is a scalar while a fluxgate can measure a vector. Compared to unshielded high-temperature SQUID, fluxgates may have a similar noise level, but their measurement range is much wider (Paperno, 2004). GMI and GMR have low precision and high noise. The fluxgate magnetometers are easy to use, have small volumes, and are inexpensive. Fluxgate sensors also have high resolutions (up to $10^{-11}$ T) and wide magnetic field measurement ranges (below $10^{-8}$ T), which are suitable for using with CSEM methods. The Bartington instruments Mag-03MSESL (the United Kingdom) was ideal for our borehole EMR unit. Its noise level is below 6 pT/√Hz at 1 Hz. In addition, it can be powered from any ±12 V supply and its outputs are in the form of three analog voltages from 0 to ±10 V, proportional to Bx, By, and Bz. The bandwidth is DC~3 kHz. A square sensor allows for easy setup in the field and offers a better axis alignment error, and a measuring range of ±100 µT allows the Earth's entire magnetic field to be measured.

### 2.2 Acquisition circuit

The acquisition circuit contains amplifiers, filters and analog-to-digital conversions used to acquire the Ez component and the orthogonal three-axis magnetic field signals.

Since the type and range of the electrode output signals and the magnetometer output signals are different, there are some differences in the principle of the electric field signal and magnetic field signal acquisition channels. The electric field signal originates from the potential difference between the two electrodes and belongs to the differential signal. Due to the size limitation of the probe, the dipole length between the two electrodes is short, and the signal input of the channel is extremely weak, with an input range of ±4 mVpp~±250 mVpp. Therefore, in order to obtain a signal with a signal-to-noise ratio that satisfies the required signal, the channel signal is amplified by a pre-amplifier and a programmable amplifier with a 10x gain. The low-pass network is placed at the input to the instrumentation amplifier for filtering, filtering out the common-mode signals as much as possible, preserving the differential-mode signals, and increasing the common-mode rejection ratio. The magnetic field signal is simply attenuated because the fluxgate sensor is directly converted to a single-ended voltage signal output, with a range of ±10 V.

Using electrical isolation can effectively reduce the interference between the analog and digital circuits. The acquisition circuit is connected to the control circuit through an isolated gate driver to ensure the proper transmission of clock signals,


configuration commands, and data. The output stages of the four analog-to-digital converter (ADC) are daisy-chained, effectively reducing the use of line connections and isolation circuitry. The FPGA provides synchronization for simultaneous
multi-channel data acquisition.

## 2.3 Control circuit

The control circuit contains the following parts: the FPGA, ARM, OCXO, real-time clock (RTC), and a secure digital memory card (SD). The control circuit uses a combination of the ARM and FPGA to implement packing, reading, storing, transmitting, collecting circuit, and basic data status extraction.
The FPGA (EP4CE22F17C8N from Intel, the United States) is the bridge between the ARM and the analog acquisition circuits. It is directly connected to the acquisition circuit through an IO interface to control the logic and timing relationships. It provides the main clock and synchronization signals for the Analog-to-Digital Converter (ADS1271 from Analog device, the United States), configures the sampling rate and gain, receives the serial data sent by the acquisition circuit, and packages and sends the data to the ARM controller. In addition, through the spread spectrum clock (SSC) bus, the serial peripheral interface
(SPI) bus and the parallel IO interface complete the bi-directional communication with the ARM, send data to the ARM through the SSC bus, receive configuration commands sent by the ARM through the SPI bus, and complete the acquisition circuit.

The ARM uses an AT91SAM9G45 (Atmel, the United States) industrial grade processor. This processor is based on the ARM 926EJ-S, and adopts the Harvard architecture. The instructions and data belong to different buses, which can be
processed in parallel. The working frequency extends to 400 MHz. As part of the borehole EMR controller, the ARM mainly performs the following functions: (1) time keeping; (2) data acquisition and storing data on the SD card; and (3) communication with the wellhead unit.

## 2.4 Time synchronization circuit

In the exploration process, there are multiple acquisition locations, which means that multiple receivers work
together. The transmitter and receiver adopt a distributed design in the borehole electromagnetic detection system, so it is necessary to synchronize timing between each borehole EMR and its transmitter. Figure 4 shows a block diagram of the time synchronization technology used. We used the GPS and OCXO as the synchronization reference signals to complete time synchronization. As time changes, there is a small amount of change in the clock frequency. We use the MCU (AT91SAM9G45) to control the Digital-to-Analog Converter (DAC) to feed back a voltage amount for correction, which
meets the clock accuracy requirements for borehole electromagnetic measurements. After each borehole EMR has been synchronized, then the relative time error between the units depends only on the clock stability of the crystal used inside the borehole EMR. The current high-precision temperature-compensated crystal oscillator has a clock stability of 10-8 s / s. By testing this, the time error is less than 10 μs. In order to monitor the signal acquisition status of the electromagnetic receiver in the borehole and observe whether there is data loss, we set a timestamp to write the synchronous clock
information to a fixed length block of data. One method is for our synchronous clock to use the 24 kHz clock signal for continuous counting, which is read every time a fixed length of electromagnetic field data is acquired. Time



synchronization technology ensures that the borehole EMR time and reference GPS signal remain accurate and consistent, providing a uniform time coordinate for data post-processing.

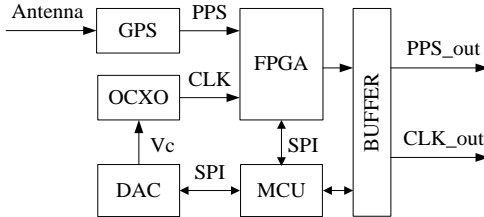

**Figure 4: Block diagram of the time synchronization technology.**

## 2.5 Attitude circuit

The attitude circuit measures the attitude of the electromagnetic receiver in the borehole. In terrestrial studies, the fluxgate can be placed in a specified direction on the ground, but this is difficult to observe in a borehole. Therefore, when measuring the three-axis magnetic field signal in real time in the borehole, it is also necessary to measure the
155 attitude angle of the borehole EMR in order to determine attitude information. The size and direction of the magnetic field are determined using the coordinate transformation between the instrument coordinate system and the geodetic coordinate system. Figure 5 shows a block diagram of the hardware in the attitude circuit. The three-axis accelerometer and the three-axis magnetometer are used to obtain the voltage signal. The analog signal is amplified and converted to digital using a signal conditioning circuit. The converted digital signal is angularly converted by the MCU (80C51F320)
to obtain its parameters, and finally the attitude information is output by the serial port. During the acquisition process, the attitude circuit returns angle information by receiving the data request command, and the attitude information is read once for each fixed length data block. The azimuth measurement range is 0°~360°, and the apex angle measurement range is 0°~180°. One notable issue here is that, in order to obtain an accurate angle between the three-axes of the fluxgate sensor and the field source, high-precision measurement of the angle is required, and the attitude measurement
also needs to measure the axial direction. The fluxgate measurement axis remains at the same height.

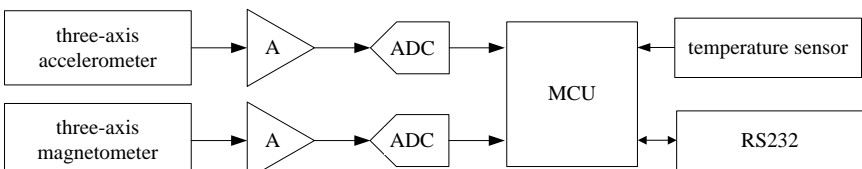

**Figure 5: Block diagram of the hardware of the attitude circuit.**

## 2.6 Communication circuit

The long distance communication circuit completes the communication between the borehole probe and the
170 wellhead unit to control the borehole probe, its data transmission, and display its status. The general drilling depth is





2000 meters, which adds a long-distance communication problem.

There are available communication solutions for logging instruments, including single core cable communication, optical fiber communication, CAN bus network, RS485, RS422, among others. Single-core cable communication uses coded signals, which increases the complexity of the hardware circuit. Optical fiber communication has the advantage of fast transmission speed and large transmission bandwidth, but the cost is high. Compared with CAN bus network transmission, RS485 and RS422 can transmit over larger distances. Both are differential transmissions, which are good at suppressing interference in the transmission line. However, RS422 is a full-duplex mode and RS485 is a half-duplex mode. Our proposed receiver used a built-in lithium battery and local data storage. The borehole EMR itself has large-capacity data storage and high-speed data throughput, thus it does not need to use a cable power supply and a cable to transmit high capacity data. Only simple control commands and basic status are transmitted through the cable. The rate of signal transmission is not high, approximately 200 bytes per second. Our comparison found that the four-core cable RS422 communication scheme has advantages of low power consumption, strong anti-electromagnetic interference capability and long communication distance, so it was selected for our unit. When the module runs at a baud rate of 9600 bps, the longest transmission distance reaches 1800 m.

In addition, the RJ-45 interface was configured to meet the high-speed data export through the network cable. When the borehole EMR is on the ground, they can communicate over Ethernet, with a transmission rate of up to 10 MB/s.

## 3 Performance test of the borehole EMR

To test the function, performance, and stability of the borehole EMR, a series of laboratory experiments and field tests were carried out. Laboratory testing focused on the noise floor, bandwidth, non-linearity error, etc. According to the actual exploration needs of the mining area, the field test carried out CSEM data collection work in a borehole to evaluate the performance of the borehole EMR system.

### 3.1 Laboratory tests

#### 3.1.1 Noise floor

In order to determine the noise contribution from the borehole EMR itself, the magnetic field channel input of the receiver was shorted to ground while the electrical field channel differential input was shorted. We mainly observed the noise floor level in a frequency range of DC -10 kHz. Following Fast Fourier Transformation (FFT) calculation, the noise power spectrum density (PSD) shown in Figure 6 was obtained. Figure 6 shows that the noise floor of the three magnetic field channels was lower than 300 nV/√Hz at 1 kHz, and the electric field channel noise floor was approximately 10 nV/√Hz at 1 kHz. The fluxgate sensitivity was 100 μV/nT, which results in magnetic field channel noise less than 3pT/√Hz at 1 kHz. There were almost no peaks in the pass band, except for high interference levels at 50 Hz and 150 Hz.



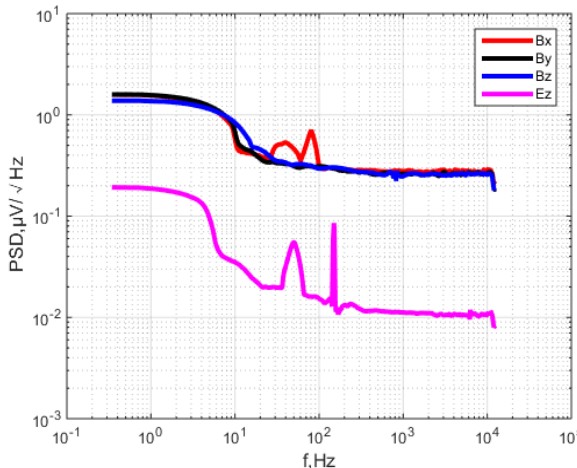

**Figure 6: Noise power spectral density. The pink curve represents the electric field channel. The red, black and blue curves represent the magnetic field channels $B_x$, $B_y$, and $B_z$, respectively.**

### 3.1.2 Bandwidth

Theoretically, the frequency component of the unit impulse function is uniformly distributed over the entire frequency range, that is, the frequency component ranges from zero to infinity. According to this principle, we used a narrow pulse signal to approximate the unit impulse signal in order to complete the acquisition channel bandwidth test. The standard signal generator outputs a pulse signal with a pulse width of 20.83 μs, which are connected to the electrical and magnetic channels, respectively. The pulse signal was continuously acquired for 1 min. Then, the acquired data were subjected to FFT analyses. Test results showed that -3 dB bandwidth for each channel can reach DC~10 kHz.

### 3.1.3 Non-linearity error

We also tested the non-linearity of the borehole EMR that is caused by the signal conditioning circuit and the analog to digital converter in the acquisition circuit. The direct-current voltage calibration source(Fluke 5720)outputs a series of direct-current voltages, which the receiver receives one by one. The fitted line of the collected voltage values was obtained by least squares fitting, and the deviation between the fitted straight line and the standard voltage curve was calculated to obtain the non-linear error of the borehole EMR. Since the borehole EMR has different electric and magnetic channel input ranges, we measured them separately. For the electric channel, the high-pass part of the channel band-pass filter was removed and the output direct-current voltage ranged from -250 mV to 250 mV. The standard voltage value of 20 mV was equally spaced. This result was calculated to have a non-linear error of 33 ppm, as shown in Figure 7(a). For the magnetic channels, the output ranged between -10 V~+10 V, and had a standard voltage value of 2 V equal intervals. These results are shown in Figure 7(b), and the channel-to-channel consistency is good. The three-channel non-linearity errors for the magnetic channels were 6.5 ppm, 7.8 ppm, and 8.5 ppm, respectively.


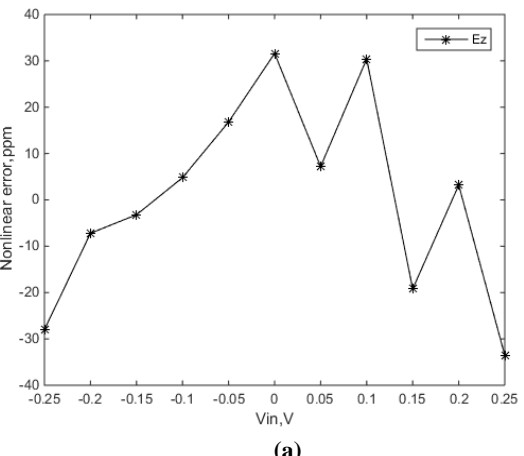
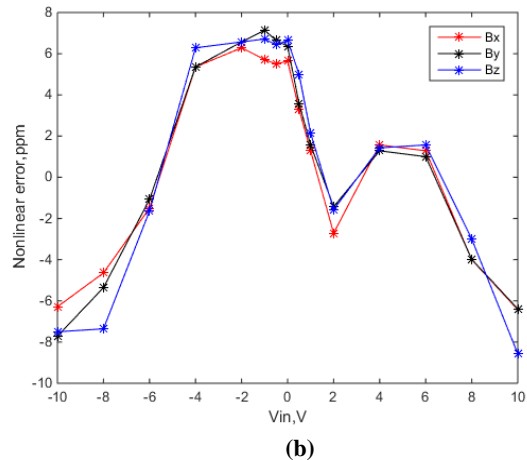

|  |  |
| :---: | :---: |
| (a) | (b) |

**Figure 7: Non-linear errors of channel fitting straight line and the standard voltage curve. (a) Electrical component (b) Magnetic component**

### 3.2 Field tests

Field observations were made to evaluate the field performance of our newly developed borehole EMR. The mine test site is located in Linxi County, Chifeng City, in the Inner Mongolia Autonomous Region, at approximately 43°N, 118°E, at an altitude of ~1 km. Figure 8 shows the field layout of the experiment. To ensure a sufficient signal-to-noise ratio, the transmitter (EMT48) was located 2.4 km south of the test point. The two (A, B) transmitting electrodes were 1.3 km apart. EMT48 is a multi-function borehole ground electromagnetic transmitting system (Wang, et al., 2018), from which the maximum power was output continuously for over 8 h, with a power output of more than 48 kW at a current above 60 A. According to the CSAMT method, the signal covering the frequency band of 0.9375 Hz~9600 Hz was transmitted. One launch circle period lasted 50 min and the maximum current was 50 A. We arranged two borehole EMRs including BH1 and BH2, for taking simultaneous measurements. The sampling rate was switched according to the transmission frequency, and was set to 24 kHz, 2.4 kHz, 150 Hz, and 15 Hz.

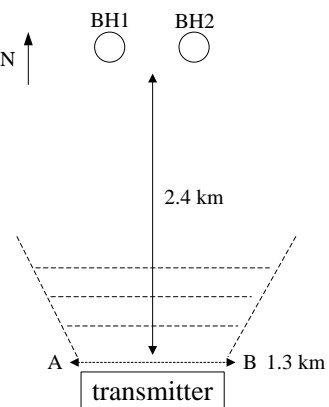





**Figure 8: Field layout of the experiment.**

First, by performing FFT analyses of all 41 frequency points, we mainly observe the signals of the $E_z$, $B_x$, $B_y$, and $B_z$
components during a launch cycle. Figure 9 shows the signal time-frequency spectra from BH1 and BH2. The expected
target frequency can be seen more intuitively from the time-frequency spectrum. The emission source was a horizontal
electrical dipole source. The two source components, $B_x$ and $B_y$, of the horizontal source are clearly observed. The $E_z$ and
$B_z$ components are perpendicular to the field source, the received signal is relatively weak, and the $E_z$ is easily observed,
compared to the $B_z$ component. In the low frequency band, the source signal is strong, the signal-to-noise ratio is high,
and the frequency signal below 200 Hz is more obvious. As the frequency increases, the signal amplitude attenuation
increases and the signal-to-noise ratio decreases. We found that, when the frequency of the transmitted signal reached
960 Hz, the acquired signal was attenuated to 6 dB, and signals larger than 1 kHz were basically unobservable in the
frequency domain.

(a) Channel BH1 signal ($B_x$ component)    (b) Channel BH1 signal ($B_y$ component)

(c) Channel BH1 signal($E_z$ component)    (d) Channel BH2 signal ($B_x$ component)



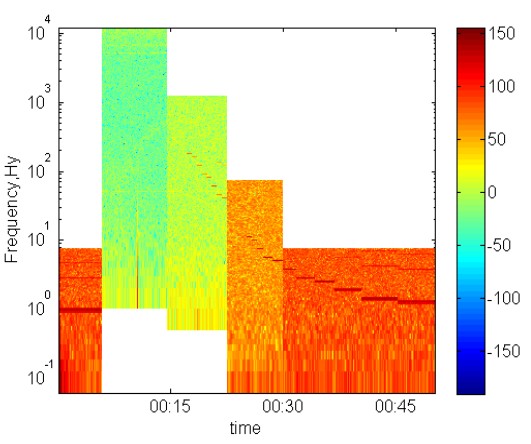
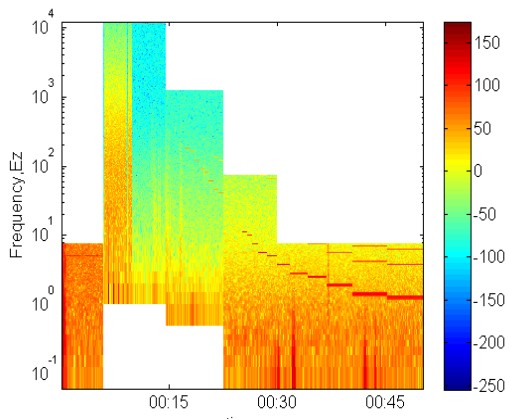

**(e) Channel BH2 signal (B_y component)**     **(f) Channel BH2 signal (E_z component)**

**Figure 9: Time-frequency spectra.**

When comparing the borehole EMR system proposed in this paper with the performance indicators of the same type of currently available products, as shown in Table 1, existing receivers only measure magnetic field signal information, but the borehole EMR mentioned in this paper innovatively adds electric field signals and synchronously acquires four channels of data, which makes the results more complete and reliable. We observed that our system has obvious advantages in bandwidth, where the highest frequency can reach 10 kHz. Variable sampling rate sampling that is

dependent on the transmission frequency is also a key point.

**Table 1. Comparison of the performance indicators of available borehole electromagnetic observation equipment**

| Model | DigiAtlantis Probe | PEM | This study |
|---|---|---|---|
| Exploration method | TDEM | TDEM | CSEM |
| Sensor | Fluxgate | Induction coil, Fluxgate | Electrode, Fluxgate |
| Number of channels | 3 | 3 | 4 |
| Bandwidth | DC - 4 kHz | | DC - 3kHz (magnetic field) DC - 10 kHz (electric field) |
| Resolution | 24 bit | 26 bit | 24 bit |
| Dynamic Range | | 157 dB | 110 dB (magnetic field) 103dB (electric field) |
| Sampling Rate | 25 kHz | 250 kHz | 24 kHz, 2.4 kHz, 150 Hz, 15 Hz |
| Power supply | Lithium battery | Rechargeable nickel-cadmium battery | Lithium battery |

## 4 Conclusions

    According to the measurement requirements of the borehole-surface electromagnetic method, we have developed a borehole EMR to further advance the development of this method. Borehole EMR has the capability to record up to four

channels of data simultaneously, including the electric and magnetic field signals of Ez, Bx, By, and Bz. The borehole EMR increases the collection of electric field signals, effectively improves the accuracy of data interpretation, and



provides technical support for multi-component observations. Our design using a low noise data collector achieved magnetic field noise less than 6pT/√Hz at 1 kHz and the electric field noise floor was approximately 10 nV/√Hz at 1 kHz. The -3 dB bandwidth of the electric field reached DC~10 kHz while the magnetic field reached DC~3 kHz. Field observations were carried out to evaluate the borehole EMR system's field performance. All borehole EMRs completed data collection of their acquired data and effectively verified the full waveform synchronous recording of four channels and time synchronization. Results of the experiments show that our system functioned well and that high quality CSEM signals were obtained.

**Data Availability**

The raw data of experiment are available upon request (2010180031@cugb.edu.cn).

**Author Contributions**

SXS and MEA designed and tested electronics. SJ is the project leader. MD and KC provide idea and guidance, including the experiment. SXS prepared the manuscript with contributions from all co-authors.

**Competing interests**

The authors declare that they have no conflict of interest.

**Acknowledgements**

Special thanks to the general funding provided by "13th Five-Year" National Key R&D Program (2017YFF0105704) and "12th Five-Year" National 863 Program (2014AA06A603).

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
