# Peer review of "A new borehole electromagnetic receiver developed for controlled source electromagnetic methods"

_Geoscientific Instrumentation, Methods and Data Systems, 2019_

## Referee Comment (RC1) · Anonymous Referee #1 · 7 Feb 2020

The authors should clarify in a clear way what is the real novelty of this contribution and what is the real step forward with respect to the pertinent literature. At this stage, this is not clearly stated. The work sounds as a technical report and has not a scientific soundness.

About the specific comments.

1) The introduction is written in a bad and generic way. There are few statements wrong such as "The borehole-surface electromagnetic method is an electromagnetic survey method that supplies a high-power alternating current with a horizontal electrical dipole and receives an electromagnetic response from the ground, tunnel, or borehole being measured." This definition of the electromagnetic method is not correct since the description of the sensing phenomenon is nots correctly stated.

[Figure]

Other statements are provided in a generic way and without references, such as . "In comparison with the conventional surface electromagnetic method, the borehole-surface electromagnetic method has a deeper detection depth and a higher resolution." "Multi-component data can help researchers better interpret the relevant properties of subsurface media."

2) What is the attitude information ? Please, provide the explicit definition/meaning of "attitude".

3) Row:200. What is the nature of the interference at 50 and 200 Hz ? Why the double interference for Bx ?

4) Row 205. Please show the spectrum of the signal.

5) Figure 8 is not clear and should be redone.

6) Row 240. "The expected target frequency can be seen more intuitively from the time-frequency spectrum." has not scientific meaning.

7) Figure 9. Is the scale in the figures provided in dB ? what is the unit of the time (hours, minutes, seconds..) ?

8) Explain explicitly the details on how the frequency time analysis has been done. At this stage, the section "field tests" is written in a very bad way.

---

## Short Comment (SC1) · 5 Mar 2020

Dear referee, I sincerely thank you for your suggestions. My answers to your questions are listed below.

1) The introduction is written in a bad and generic way. There are few statements wrong such as "The borehole-surface electromagnetic method is an electromagnetic survey method that supplies a high-power alternating current with a horizontal electrical dipole and receives an electromagnetic response from the ground, tunnel, or borehole being measured." This definition of the electromagnetic method is not correct since the description of the sensing phenomenon is nots correctly stated. Other statements are provided in a generic way and without references, such as. "In comparison with the

conventional surface electromagnetic method, the borehole surface electromagnetic method has a deeper detection depth and a higher resolution." "Multi-component data can help researchers better interpret the relevant properties of subsurface media."

Response: Thank you for your comment. Please note, in the revised manuscript, the following modifications have been made to the sections pointed out by you. "The borehole-surface electromagnetic method is an electromagnetic survey method that can deliver high-power alternating current with different frequencies through horizontal electric dipoles, and receive three-dimensional electromagnetic signals from the ground, tunnels, or boreholes." (p. 1, lines 23-25) "Compared to the conventional surface electromagnetic method, the borehole surface electromagnetic method has a deeper detection depth and a higher resolution (Li T.T. et al., 2013)." (p. 1, lines 25-26) "Multi-component data can help researchers better interpret the relevant properties of subsurface media (Duncan et al., 1998)." (p. 2, line 35)

2) What is the attitude information? Please, provide the explicit definition/meaning of "attitude".

Response: Thank you for your comment. As suggested, the term attitude information has been defined in the revised manuscript as follows. "The attitude information includes pitch, roll, and yaw angles." (p. 6, lines 153-154)

3) Row: 200. What is the nature of the interference at 50 and 200 Hz? Why the double interference for Bx?

Response: Thank you for your comment. The nature of the interference is the power frequency interference and its harmonic interference. "There were almost no peaks in the pass band, except for high interference levels at 50 Hz and 150 Hz)." (p. 7, lines 200-201)

4) Row 205. Please show the spectrum of the signal.

Response: Thank you for your comment. As suggested, the spectrum of the signal has

been explained in the revised manuscript on (p. 8, line 206).

5) Figure 8 is not clear and should be redone.

Response: Thank you for your comment. As advised, we have revised Figure 8 to read as follows.

Figure 9: Field layout of the experiment. (p. 10, lines 239-240) BH1 and BH2 are placed at a certain depth in the borehole. The transmitter and the electrodes are on the ground.

6) Row 240. "The expected target frequency can be seen more intuitively from the time-frequency spectrum." has not scientific meaning.

Response: Thank you for your comment. The text has been revised as follows. "Figure 10 shows the time-frequency spectrum of the signals from BH1 and BH2, from which the expected target frequency can be seen more clearly." (p. 10, lines 242-243)

7) Figure 9. Is the scale in the figures provided in dB? what is the unit of the time (hours, minutes, seconds..) ?

Response: Thank you for your comment. We have provided the following explanation in the revised manuscript to address your feedback. "The scale in the figures is provided in dB; however, the data in the figure is calculated with the formula 10logX. The units of the time and frequency are minutes and Hz, respectively." (p. 11, lines 250-251)

8) Explain explicitly the details on how the frequency time analysis has been done. At this stage, the section "field tests" is written in a very bad way.

Response: Thank you for your comment. The time window is used to perform a periodic sparse fast Fourier transform on the signal to obtain the time-frequency spectrum of the signal.
* * *
[Figure]

https://doi.org/10.5194/gi-2019-37, 2020.

[Figure]

**Fig. 1.**

[Figure]

**Fig. 2.**

---

## Referee Comment (RC2) · Anonymous Referee #2 · 26 May 2020

This is an excellent piece of work, developing new instrumentation for measurement of EM fields in boreholes for controlled-source EM surveys. The description of the instrumentation is detailed and comprehensive; the tests are convincing.

---

## Short Comment (SC2) · 28 May 2020

Dear referee,

Thanks for your comments and looking forward to receiving your suggestions again.
* * *

---

## Author Comment (AC1) · 21 Jun 2020

Dear Editors, We thank you and the reviewers for your thoughtful suggestions and insights. The responses to all comments have been prepared and attached herewith.

Anonymous Referee #1 Dear referee, I sincerely thank you for your suggestions. My answers to your questions are listed below.

1) The introduction is written in a bad and generic way. There are few statements wrong such as "The borehole-surface electromagnetic method is an electromagnetic survey method that supplies a high-power alternating current with a horizontal electrical dipole and receives an electromagnetic response from the ground, tunnel, or borehole being measured." This definition of the electromagnetic method is not correct since the

[Figure]

description of the sensing phenomenon is nots correctly stated. Other statements are provided in a generic way and without references, such as. "In comparison with the conventional surface electromagnetic method, the borehole surface electromagnetic method has a deeper detection depth and a higher resolution." "Multi-component data can help researchers better interpret the relevant properties of subsurface media."

Response: Thank you for your comment. Please note, in the revised manuscript, the following modifications have been made to the sections pointed out by you. "The borehole-surface electromagnetic method is an electromagnetic survey method that can deliver high-power alternating current with different frequencies through horizontal electric dipoles, and receive three-dimensional electromagnetic signals from the ground, tunnels, or boreholes." (p. 1, lines 23-25) "Compared to the conventional surface electromagnetic method, the borehole surface electromagnetic method has a deeper detection depth and a higher resolution (Li T.T. et al., 2013)." (p. 1, lines 25-26) "Multi-component data can help researchers better interpret the relevant properties of subsurface media (Duncan et al., 1998)." (p. 2, line 35)

2) What is the attitude information? Please, provide the explicit definition/meaning of "attitude".

Response: Thank you for your comment. As suggested, the term attitude information has been defined in the revised manuscript as follows. "The attitude information includes pitch, roll, and yaw angles." (p. 6, lines 153-154)

3) Row: 200. What is the nature of the interference at 50 and 200 Hz? Why the double interference for Bx?

Response: Thank you for your comment. The nature of the interference is the power frequency interference and its harmonic interference. "There were almost no peaks in the pass band, except for high interference levels at 50 Hz and 150 Hz)." (p. 7, lines 200-201)

4) Row 205. Please show the spectrum of the signal.

Response: Thank you for your comment. As suggested, the spectrum of the signal has been explained in the revised manuscript on (p. 8, line 206).

5) Figure 8 is not clear and should be redone.

Response: Thank you for your comment. As advised, we have revised Figure 8 to read as follows.

Figure 9: Field layout of the experiment. (p. 10, lines 239-240) BH1 and BH2 are placed at a certain depth in the borehole. The transmitter and the electrodes are on the ground.

6) Row 240. "The expected target frequency can be seen more intuitively from the time-frequency spectrum." has not scientific meaning.

Response: Thank you for your comment. The text has been revised as follows. "Figure 10 shows the time-frequency spectrum of the signals from BH1 and BH2, from which the expected target frequency can be seen more clearly." (p. 10, lines 242-243)

7) Figure 9. Is the scale in the figures provided in dB? what is the unit of the time (hours, minutes, seconds..) ?

Response: Thank you for your comment. We have provided the following explanation in the revised manuscript to address your feedback. "The scale in the figures is provided in dB; however, the data in the figure is calculated with the formula 10logX. The units of the time and frequency are minutes and Hz, respectively." (p. 11, lines 250-251)

8) Explain explicitly the details on how the frequency time analysis has been done. At this stage, the section "field tests" is written in a very bad way.

Response: Thank you for your comment. The time window is used to perform a periodic sparse fast Fourier transform on the signal to obtain the time-frequency spectrum of the signal.

Anonymous Referee #2 Dear referee, Thanks for your comments and looking forward to receiving your suggestions again.

Sincerely, Sixuan Song School of Geophysics and Information Technology China University of Geosciences Beijing, China 86-18810861682 2010180031@cugb.edu.cn

Please also note the supplement to this comment:
https://gi.copernicus.org/preprints/gi-2019-37/gi-2019-37-AC1-supplement.pdf
* * *
![Magnitude vs Frequency plot showing magnitude in uV on the y-axis from 10^-1 to 10^2 and frequency in Hz on the x-axis from 10^-1 to 10^5, with a roughly flat response around 2 uV that drops off near 10^4 Hz]

**Fig. 1.**

[Figure]

**Fig. 2.**

---

## Author Response (AR1)

[July 17,2020]

Jean Dumoulin, Salvatore Grimaldi, and Håkan Svedhem
Executive Editor
*Geoscientific Instrumentation, Methods and Data Systems*

Dear Editors:

I wish to re-submit the article titled "A new borehole electromagnetic receiver developed for controlled source electromagnetic methods". The manuscript ID is 530682.

We thank you and the reviewers for your thoughtful suggestions and insights. The manuscript has benefited from these insightful suggestions. I look forward to working with you and the reviewers to move this manuscript closer to publication in the *Geoscientific Instrumentation, Methods and Data Systems*.

The manuscript has been rechecked and the necessary changes have been made in accordance with the reviewers' suggestions. The responses to all comments have been prepared and attached herewith.

1) The introduction is written in a bad and generic way. There are few statements wrong such as "The borehole-surface electromagnetic method is an electromagnetic survey method that supplies a high-power alternating current with a horizontal electrical dipole and receives an electromagnetic response from the ground, tunnel, or borehole being measured." This definition of the electromagnetic method is not correct since the description of the sensing phenomenon is nots correctly stated.
Other statements are provided in a generic way and without references, such as. "In comparison with the conventional surface electromagnetic method, the borehole surface electromagnetic method has a deeper detection depth and a higher resolution." "Multi-component data can help researchers better interpret the relevant properties of subsurface media."

Response: Thank you for your comment. Please note, in the revised manuscript, the following modifications have been made to the sections pointed out by you.
 "The borehole-surface electromagnetic method is an electromagnetic survey method that can deliver high-power alternating current with different frequencies through horizontal electric dipoles, and receive three-dimensional electromagnetic signals from the ground, tunnels, or boreholes." (p. 1, lines 23-25)
 " Compared to the conventional surface electromagnetic method, the borehole surface electromagnetic method has a deeper detection depth and a higher resolution (Li T.T. et al., 2013)." (p. 1, lines 25-26)
 "Multi-component data can help researchers better interpret the relevant properties of subsurface media (Duncan et al., 1998)." (p. 2, line 34-35)

2) What is the attitude information? Please, provide the explicit definition/meaning of "attitude".

Response: Thank you for your comment. As suggested, the term attitude information has been defined in the revised manuscript as follows.
 "The attitude information includes pitch, roll, and yaw angles." (p. 6, lines 150-151)

3) Row: 200. What is the nature of the interference at 50 and 200 Hz? Why the double interference for Bx?

Response: Thank you for your comment. The nature of the interference is the power frequency interference and its harmonic interference. "There were almost no peaks in the pass band, except for high interference levels at 50 Hz and 150 Hz)." (p. 8, lines 196-197)

4) Row 205. Please show the spectrum of the signal.

Response: Thank you for your comment. As suggested, the spectrum of the signal has been explained in the revised manuscript on (p. 9, line 202-203).

5) Figure 8 is not clear and should be redone.

Response: Thank you for your comment. As advised, we have revised Figure 8 to read as follows.

Figure 9: Field layout of the experiment. (p. 11, lines 235-237) BH1 and BH2 are placed at a certain depth in the borehole. The transmitter and the electrodes are on the ground.

6) Row 240. "The expected target frequency can be seen more intuitively from the time-frequency spectrum." has not scientific meaning.

Response: Thank you for your comment. The text has been revised as follows. "Figure 10 shows the time-frequency spectrum of the signals from BH1 and BH2, from which the expected target frequency can be seen more clearly." (p. 11, lines 239-240)

7) Figure 9. Is the scale in the figures provided in dB? what is the unit of the time (hours, minutes, seconds..) ?

Response: Thank you for your comment. We have provided the following explanation in the revised manuscript to address your feedback. "The scale in the figures is provided in dB; however, the data in the figure is calculated with the formula 10logX. The units of the time and frequency are minutes and Hz, respectively." (p. 12, lines 247-248)

8) Explain explicitly the details on how the frequency time analysis has been done. At this stage, the section "field tests" is written in a very bad way.

Response: Thank you for your comment. The time window is used to perform a periodic sparse fast Fourier transform on the signal to obtain the time-frequency spectrum of the signal.

Thank you for your consideration. I look forward to hearing from you.

Sincerely,
Sixuan Song
School of Geophysics and Information Technology
China University of Geosciences
Beijing, China
86-18810861682
2010180031@cugb.edu.cn

---

## Editor Decision (ED1)

Abstract:

10nV/sqrt (Hz) is stated. This is a potential not a field strength, while the discussion is about electric fields. Please explain why this is chosen, or correct.

The expression "fluxgate" is a short form derived from "fluxgate magnetometer". Please use the full form at least the first time this is used. (in the abstract and in the main text)

"Results of our experiments support the claim that high-quality CSEM signals can be obtained using this new borehole electromagnetic receiver, and that the electric field component exhibits sufficient advantages for measuring the vertical component of the electric field." – This is not discussed, please describe what is sufficient and for what purpose. What is "high-quality CSEM signals"? What is "sufficient advantage"?

Last sentence in paragraph 1 states "The receiver realizes high-precision acquisition of the three-axis magnetic field components and the vertical electric field component in the borehole, with broad bandwidth and large dynamic ranges, and stores and transmits status data that contains the root mean squares (RMS) of the magnetic and electric field signals, attitude, orientation, depth, and temperature." What is the difference between attitude and orientation?

Paragraph2: It is stated " …. uses a global positioning system (GPS) and…". Please explain how (if) the GPS unit will work underground (in the borehole).

Paragraph 2.2 "… four analog-to-digital converters(ADCs) are daisy-chained". In Fig. 3 it appears that the ADCs are individually connected to the FPGA (in parallel) and not daisy-chained. Please correct either text or drawing.

Paragraph 2.4 "The current high-precision temperature-compensated crystal oscillator… ". This oscillator is not shown in the diagram. Before this only an OCXO has been discussed. Is this a second oscillator? Please explain. If there are two clocks, why is not the OCXO used as a master oscillator but only as a synchronisation device?

Paragraph 2.5 "In terrestrial studies…". All studies discussed here are terrestrial studies. Supposedly you mean surface studies, or similar. Please correct.

Paragraph 3.1.3. Please explain why receiver nonlinearity was measured at DC while the real measurements will be made at AC.

Paragraph 3.2 and Fig 9. It is not clear how x, y, and z are oriented. Is Bz and Ez the vertical direction? If so, then it is clear that the Ez signal from the horizontally oriented transmitter is low. But the Bz should have a significant magnetic flied component. Why is this not the case? Which direction in the receiver is parallel to the transmit dipole A-B (where the current is injected)? That B field measurement should be low, not the vertical B Field. Please explain.

Fig.9 At what depth were the BH1 and BH2 instruments located?

Fig.10. For all panels:  What is time? mm:ss or hh:mm?   Y-axis is labelled as Frequency,Hy or Frequency,Ez and similar. Better write Frequency, Hz  The component is already given under each panel.

It is stated that 41 frequency steps are made. That many cannot be seen. It appears that the duration of the weak high frequency steps are very short, while the duration of the strong low frequency steps are much longer. Why is this? It seems that selecting the opposite would be more logical.

Fig 10. Panels are not well aligned, please organise such that corresponding panels come above each other (or next to each other), f. ex. :

Bx1  By1  Ez1

Bx2  By2  Ez2

or:

Bx1  Bx2

By1  By2

Ez1  Ez2

The caption states: "The scale in the figures is provided in dB; however, the data in the figure is calculated with the formula 10logX."    Then the scale is not dB as dB always refers to a ratio of power. What is the unit of X?  i.e. what is the reference in the  10LogX ?  V/m for E and nT for B?

Paragraph 3 (end): It is stated: "We observed that our system has obvious advantages in bandwidth, where the highest frequency can reach 10 kHz". It is not clear why this is an advantage. In Fig 10 one can see that no signals exist above a few hundred Hz. Please explain. The following sentence has the word "sampling" repeated, making the sentence difficult to understand. (typo?)

Paragraph 4.  It is stated:" According to the measurement requirements of the borehole surface electromagnetic method… "  Such requirements have not been discussed or even mentioned in this paper. Thus, no conclusions can be drawn on that. Please add a discussion in the main text or remove this sentence.  As in the abstract, here again the Electric field is given as a potential (10nV/sqrt(Hz)). Please correct.

---

## Author Response (AR2)

[December 16, 2020]

Håkan Svedhem
Executive Editor
Geoscientific Instrumentation, Methods and Data Systems

Dear Editors:

I sincerely thank you for your suggestions. My answers to your questions are listed below.

**1. 10nV/sqrt (Hz) is stated. This is a potential not a field strength, while the discussion is about electric fields. Please explain why this is chosen, or correct.**

**Response:** The dipole length between the two electrodes was 50 cm, so the electric field noise floor was approximately 20 nV·m$^{-1}$/√Hz at 1 kHz. As suggested, the text has been revised as follows: "The receiver achieved a magnetic field noise of less than 6 pT/√Hz at 1 kHz, and the electric field noise floor was approximately 20 nV·m-1/√Hz at 1 kHz."

**2. The expression "fluxgate" is a short form derived from "fluxgate magnetometer". Please use the full form at least the first time this is used. ( in the abstract and in the main text)**

**Response:** As suggested, the expression "fluxgate" has been defined in the first instance of its occurrence in the revised manuscript.

**3. "Results of our experiments support the claim that high-quality CSEM signals can be obtained using this new borehole electromagnetic receiver, and that the electric field component exhibits sufficient advantages for measuring the vertical component of the electric field." – This is not discussed, please describe what is sufficient and for what purpose. What is "high-quality CSEM signals"? What is "sufficient advantage"?**

**Response:** "High-quality CSEM signals" represents a signal with a high signal-to-noise ratio. "Sufficient advantage" indicates that the vertical component can measure both electric and magnetic fields.

**4. Last sentence in paragraph 1 states "The receiver realizes high-precision acquisition of the three-axis magnetic field components and the vertical electric field component in the borehole, with broad bandwidth and large dynamic ranges, and stores and transmits status data that contains the root mean squares (RMS) of the magnetic and electric field signals, attitude, orientation, depth, and temperature." What is the difference between attitude and orientation?**

**Response:** The attitude information includes pitch, roll, and yaw angles, and the orientation generally means yaw angles. As suggested, the text has been revised as follows: "The receiver ensures high-precision acquisition of the three-axis magnetic field components and the vertical electric field component in the borehole, with broad bandwidth and large dynamic ranges. It also stores and transmits status data that contains the root mean squares (RMS) of the magnetic and electric field signals, attitude, depth, and temperature."

**5. Paragraph2: It is stated "…. uses a global positioning system (GPS) and…" Please explain how (if) the GPS unit will work underground (in the borehole).**

**Response:** The GPS unit synchronizes time between the borehole EMRs and the transmitter on the ground and then places it in the borehole.

**6. Paragraph 2.2 "… four analog-to-digital converters (ADCs) are daisy-chained". In Fig. 3 it appears that the ADCs are individually connected to the FPGA (in parallel) and not daisy-chained. Please correct either text or drawing.**

**Response:** As advised, we have revised Figure 3 to read as follows:

[Figure]

**7. Paragraph 2.4 "The current high-precision temperature-compensated crystal oscillator… ". This oscillator is not shown in the diagram. Before this only an OCXO has been discussed. Is this a second oscillator? Please explain. If there are two clocks, why is not the OCXO used as a master oscillator but only as a synchronisation device?**

**Response:** Please note, in the revised manuscript, the following modifications have been made to the relevant sections: "The current oven-controlled crystal oscillator has a clock stability of 10-8 s s-1; testing revealed that the time error is less than 10 μs." There is only one clock, which is an oven-controlled crystal oscillator (OCXO).

**8. Paragraph 2.5 "In terrestrial studies…" All studies discussed here are**

**terrestrial studies. Supposedly you mean surface studies, or similar. Please correct.**

**Response:** The text has been revised as follows: "In surface studies, the fluxgate can be placed in a specific direction on the ground; however, this is difficult to achieve in a borehole."

**9. Paragraph 3.1.3. Please explain why receiver nonlinearity was measured at DC while the real measurements will be made at AC.**

**Response:** The non-linearity test is mainly to evaluate equipment performance. The non-linearity error will also influence the real measurements that will be made at AC.

**10. Paragraph 3.2 and Fig 9. It is not clear how x, y, and z are oriented. Is Bz and Ez the vertical direction? If so, then it is clear that the Ez signal from the horizontally oriented transmitter is low. But the Bz should have a significant magnetic flied component. Why is this not the case? Which direction in the receiver is parallel to the transmit dipole A-B (where the current is injected)? That B field measurement should be low, not the vertical B Field. Please explain.**

**Response:** Before placing the borehole EMR in the field test, Bz and Ez were in the vertical direction, but during the movement into the borehole, the borehole EMR attitude changed randomly. Based on your question, we checked the original data. Due to an oversight, the picture channel was incorrectly labeled, and I have made the necessary corrections in the resubmitted manuscript.

**11. Fig.9 At what depth were the BH1 and BH2 instruments located?**

**Response:** The BH1 and BH2 instruments were located approximately 5 m below the ground.

**12. Fig.10. For all panels: What is time? mm:ss or hh:mm? Y-axis is labelled as Frequency, Hy or Frequency, Ez and similar. Better write Frequency, Hz The component is already given under each panel.**

**Response:** The form of the time is hh:mm. The Y-axis label has been unified with the component given under each panel.

**13. It is stated that 41 frequency steps are made. That many cannot be seen. It appears that the duration of the weak high frequency steps are very short, while the duration of the strong low frequency steps are much longer. Why is this? It seems that selecting the opposite would be more logical.**

**Response:** The reason for these durations is because FFT analysis requires multiple waveform cycles to ensure accurate calculations. Thus, the durations of the weak highfrequency steps are very short, whereas the durations of the strong low-frequency steps are significantly longer.

**14. Fig 10. Panels are not well aligned, please organise such that corresponding panels come above each other (or next to each other), f. ex.:**
**Bx1 By1 Ez1**
**Bx2 By2 Ez2**
**or:**
**Bx1 Bx2**
**By1 By2**
**Ez1 Ez2**

**Response:** Thank you for your suggestion. As advised, we have revised Figure 10 in the manuscript.

**15. The caption states: "The scale in the figures is provided in dB; however, the data in the figure is calculated with the formula 10logX." Then the scale is not dB as dB always refers to a ratio of power. What is the unit of X? i.e. what is the reference in the 10LogX ? V/m for E and nT for B?**

**Response:** X represents the following formula:
$$X = |FFT[x(n) \cdot \omega(n)]|^2,$$
where $x(n)$ is the time series data, $\omega(n)$ is the window function. The result of the formula (10Log $X$)calculation is the power spectrum.

**16. Paragraph 3 (end): It is stated: "We observed that our system has obvious advantages in bandwidth, where the highest frequency can reach 10 kHz". It is not clear why this is an advantage. In Fig 10 one can see that no signals exist above a few hundred Hz. Please explain. The following sentence has the word "sampling" repeated, making the sentence difficult to understand. (typo?)**

**Response:** In the field test, the high-frequency signal attenuates more severely, so no signals exist above a few hundred Hz. Additionally, the text has been revised as follows. "The variable sampling rate that is dependent on the transmission frequency is also a key point."

**17. Paragraph 4. It is stated:" According to the measurement requirements of the borehole surface electromagnetic method… "Such requirements have not been discussed or even mentioned in this paper. Thus, no conclusions can be drawn on that. Please add a discussion in the main text or remove this sentence. As in the abstract, here again the Electric field is given as a potential (10 nV/sqrt(Hz)). Please correct.**

**Response:** As advised, we have removed this sentence "According to the measurement

requirements of the borehole surface electromagnetic method…". The test has been defined in the revised manuscript as follows: "Our design employed a low-noise data collector to achieve a magnetic field noise less than 6 pT/ √ Hz at 1 kHz, and the electric field noise floor was approximately 20 nV·m-1/ √ Hz at 1 kHz."

We sincerely hope that all our responses and clarifications are to your satisfaction. Thank you for your consideration. I look forward to hearing from you.

Sincerely,
Sixuan Song
School of Geophysics and Information Technology
China University of Geosciences
Beijing, China
86-18810861682
2010180031@cugb.edu.cn